# *Sargassum thunbergii* Extract Attenuates High-Fat Diet-Induced Obesity in Mice by Modulating AMPK Activation and the Gut Microbiota

**DOI:** 10.3390/foods11162529

**Published:** 2022-08-21

**Authors:** Dahee Kim, Jing Yan, Jinwoo Bak, Jumin Park, Heeseob Lee, Hyemee Kim

**Affiliations:** Department of Food Science and Nutrition & Kimchi Research Institute, Pusan National University, Busan 46241, Korea

**Keywords:** *Sargassum thunbergii*, anti-obesity, mice, intestinal microbiota

## Abstract

*Sargassum thunbergii* (Mertens ex Roth) Kuntze (ST) is a brown alga rich in indole-2-carboxaldehyde. This study aimed to investigate the anti-obesity effects of ethanol extract from ST in in vitro and in vivo models. In 3T3-L1 cells, ST extract significantly inhibited lipid accumulation in mature adipocytes while lowering adipogenic genes (*C/epba* and *Pparg*) and enhancing metabolic sensors (*Ampk*, *Sirt1*), thermogenic genes (*Pgc-1a*, *Ucp1*), and proteins (p-AMPK/AMPK and UCP1). During animal investigation, mice were administered a chow diet, a high-fat diet (HF), or an HF diet supplemented with ST extract (at dosages of 150 and 300 mg/kg bw per day) for 8 weeks (n = 10/group). ST extract administration decreased weight gain, white adipose tissue weight, LDL-cholesterol, and serum leptin levels while improving glucose intolerance. In addition, ST extract increased the expression of *Ampk* and *Sirt1* in adipose tissue and in the liver, as well as p-AMPK/AMPK ratio in the liver, compared to HF-fed mice. The abundance of *Bacteroides vulgatus* and *Faecalibacterium prausnitzii* in the feces increased in response to ST extract administration, although levels of *Romboutsia ilealis* decreased compared with those in HF-fed mice. ST extract could prevent obesity in HF-fed mice via the modulation of AMPK activation and gut microbiota composition.

## 1. Introduction

Obesity is a complex, chronic disease with a high risk of relapse; it is characterized by an energy imbalance in the adipose tissue in addition to chronic inflammation [1]. Obesity increases the risk of metabolic diseases such as hypertension, type 2 diabetes, and atherosclerosis; it is one of the leading causes of death and has become a cause for global concern [2]. Although there are synthetic drugs used to prevent or treat obesity, they have been associated with various side effects, including gastrointestinal issues [3]. Therefore, it is essential to discover natural compounds that are effective, safe, and nontoxic for weight loss [4].

Multiple pathways in adipocytes may be considered potential targets for anti-obesity treatment and prevention of obesity. Adipogenesis increases the number of adipocytes; C/EBPα, PPARγ, and SREBP-1c play a role in adipocyte proliferation and differentiation [5]. Lipogenesis involves the enlargement of adipocytes, and ACC and FAS are involved in the synthesis of fatty acids and triglycerides [6]. The inhibition of the expression of adipogenesis and lipogenesis markers is a potential treatment for obesity. In addition, the activation of thermogenesis, a process by which white adipocytes (WAT) transform into beige adipocytes and increase energy expenditure, has received a great deal of interest in recent obesity prevention and treatment research [7]. In contrast to WAT, which stores energy, beige and brown adipose tissues are thermogenic organs that can dissipate energy and produce heat [8]. AMPK is implicated in the development of browning adipocytes via the activation of SIRT1 and PGC-1α, as well as mitochondrial biogenesis via the induction of UCP-1 [9] and has thus become a target for weight loss.

Furthermore, the gut microbiota is being investigated as a potential target for obesity treatment or preventive strategy [10]. The gut microbiota interacts with the nutritional environment of the host, and gut dysbiosis is associated with obesity by means of causing leaky gut, LPS-induced inflammation, pro-inflammatory cytokines, and adipogenesis and lipogenesis in adipocytes, all of which contribute to the induction of inflammation, energy imbalance, and weight gain [11,12]. Dietary intervention has been documented to alter the gut microbiota composition in the short term by changing metabolic pathways [13]. High-sugar or high-fat diets increase the ratio of Firmicutes to Bacteroidetes in the gut [14], and lactic acid-producing bacteria in probiotics have been shown to contribute to adipocyte browning and thermogenesis by boosting UCP-1 expression [15]. Natural compounds have been identified as promising prebiotics for maintaining gut microbial balance, as well as preventing obesity by increasing calorie consumption, lowering adipogenesis and lipogenesis, and improving thermogenesis [16,17,18,19]. However, the precise involvement of gut microbiota in obesity and the effect of phytochemicals on the compositional balance of the gut microbiota remain unknown.

Marine seaweeds are a rich source of phytochemicals, such as polyphenols, alkaloids, tannins, and polysaccharides, and have antioxidant, anti-inflammatory, and anti-obesity properties [20,21]. Among seaweeds, the brown alga *Sargassum thunbergii* (*S. thunbergii*), indigenous to the shallow marine coasts of the northwest Pacific, has been shown to possess a number of biological functions, including anti-obesity effects [22,23,24]. *S. thunbergii* contains six indole derivatives, including indole-2-carboxaldehyde and indole-6-carboxaldehyde, which reduce lipid accumulation and adipogenesis by inhibiting SREBP-1c, PPARγ, and C/EBPα and activating AMPK in 3T3-L1 cells [25]. In addition, *S*. *thunbergii* relieves fatty liver and reduces fat accumulation in HFD-induced obese mice [26]. However, the role of *S. thunbergii* in the adipocyte browning process and composition of the gut microbiota remains unclear.

In this study, we examined the anti-obesity effects of *S. thunbergii* extract in 3T3-L1 adipocytes and in high-fat-diet-induced obese mice models. We hypothesized that a 70% ethanol extract of *S. thunbergii* would attenuate obesity by lowering adipogenesis and lipogenesis, inducing the browning of adipocytes and altering the gut microbiota composition in in vitro and in vivo obesity models.

## 2. Materials and Methods

### 2.1. Preparation of Extracts and HPLC Analysis

*S. thunbergii* (ST) extract, extracted with 70% ethanol, was acquired from the Marine Bio Bank (Seocheon-si, Korea) or ParaJeju (Jeju-si, Korea). The total phenolic and flavonoid contents of ST ethanol extract were determined spectrophotometrically using the Folin–Ciocalteu method [27] and diethylene glycol colorimetry [28]. The extract and the standard indole-2-carboxaldehyde were quantified by HPLC using Clarity^™^ chromatography software (DataApex, Prague, The Czech Republic) equipped with a C-18 column (4.6 mm × 250 mm, i.d. 5 µm) (YMC Co., Kyoto, Japan) and a DAD (70-6186A, ESA Co., Chelmsford, MA, USA) detector [29]. The wavelength was set to 290 nm. The following gradient elution process was used with solvent A (0.1% formic acid in water) and solvent B (acetonitrile): (0–10 min, 90% A and 10% B; 10–30 min, 70% A and 30% B; 30–40 min, 90% A and 10% B). LC-MS analysis was conducted using an Agilent 6530 Infinity HPLC system (Agilent Technologies, Santa Clara, CA, USA) coupled with a hybrid quadrupole time-of-flight (Q-TOF) mass spectrometer. MS signals were detected on a mass spectrometer operating in positive ionization mode. Analytes were separated on a Waters Acquity BEH C18 column (2.1 × 50 mm, 1.7 μm). The ESI method was executed using the following settings: gas temperature, 300 °C; gas flow, 9 L/min; sheath gas temperature, 350 °C; sheath flow, 11 L/min; and nebulizer pressure, 45 psig. The scan source parameters were operated with the following settings: VCap 4000 V and Fragmentor 175 V.

### 2.2. Cell Culture and Differentiation

3T3-L1 cells were obtained from the Korean Cell Line Bank (Seoul, Korea) and incubated at 37 °C with 5% CO_2_ in DMEM medium containing 2.5% HEPES, 1% penicillin/streptomycin, and 10% FBS (Hyclone, Logan, UT, USA). When cells reached 100% confluence, preadipocytes were differentiated using DMEM media containing 0.5 mM 3-isobutyl-1-methylxanthine (IBMX), 10 µg/mL insulin, and 1 µM dexamethasone (Sigma Chemical Co., St. Louis, MO, USA) for 2 days (days 1–2). The cells were subsequently maintained in culture medium containing 10 µg/mL insulin (days 3–6). Test extracts were added to the culture medium during the entire 6-day cell growth phase [29].

### 2.3. Cell Viability Assay

The viability of 3T3-L1 cells was determined using the CellTiter 96^®^Aqueous One Solution Cell Proliferation Assay (MTS; Promega, Madison, WI, USA). 3T3-L1 preadipocytes (10^4^ cells/well) were incubated overnight in 96-well plates. After 24 h, the cell medium was removed, and cells were treated for 48 h with ST extract (0–100 mg/L). Next, 10 μL of MTS reagent was added to each well, and the cells were incubated for an additional 2 h at 37 °C. The sample was then analyzed at 490 nm using a microplate reader (Spectrophotometer, Thermo Fisher, Waltham, MA, USA).

### 2.4. Oil Red O Staining

Oil Red O staining was used to detect intracellular lipid accumulation [30]. After cell differentiation, 10% formaldehyde was used to fix the cells at room temperature for 1 h. The cells were then rinsed twice with water, and 60% isopropanol was added. After 5 min of incubation, isopropanol was removed, and Oil Red O solution was added for 15 min. The plate was washed and photographed under a microscope (JP/BX-FLA, Olympus, Japan). Isopropanol was added to the plate for 10 min, and the absorbance was measured spectrophotometrically at 500 nm.

### 2.5. Animal Studies

The animal protocol was approved by the Animal Ethics Committee of the Pusan National University (PNU-2022-0130). Five-week-old male C57BL/6 mice were purchased from Orient Bio Co. (Seongnam-si, Gyeonggi-do, Korea) and maintained in a temperature-controlled environment with a 12 h light/dark cycle. After 1 week of acclimation, mice were divided into four groups (n = 10/group). The control group (Con) was fed a chow diet (5L79, Orient Bio), whereas the high-fat diet (HF) group and the experimental (ST) groups were fed a high-fat diet (D12492, Rodent Diet with 60 kcal% fat, Research Diets, New Brunswick, NJ, USA) for 8 weeks. In addition, the Con and HF groups received 100 μL of PBS daily via oral gavage, whereas the ST group received 150 (HF-StL) or 300 (HF-StH) mg/kg body weight (bw) of *S. thunbergii*. Throughout the experiment, the body weight and food intake were measured weekly. A glucose tolerance test (OGTT) was conducted 1 week before termination. Mice were fasted overnight and administered 2 g glucose/kg bw via oral gavage. Blood was drawn from the tail at 0, 30, 60, and 120 min to measure the blood glucose levels using an Accu-Chek Instant Meter (Roche Diabetes Care, Germany). One week later, the mice were euthanized, blood and feces were collected, and liver and adipose tissues were weighed and stored at −80 °C [31].

### 2.6. Histological Observations of Adipose Tissue and Liver

Adipose tissues and livers were embedded in paraffin and fixed in 4% formalin. The sections were stained with hematoxylin and eosin (H&E) and Oil Red O, and images of four distinct areas of each section were captured using an optical microscope (JP/BX-FLA, Olympus). ImageJ software (NIH, Bethesda, ML, USA) was used to compute the cross-sectional area of adipocytes.

### 2.7. Blood Parameter Analysis

The serum levels of triglycerides, total cholesterol, and high-density lipoprotein (HDL) cholesterol were determined using a kit by Asan Pharmaceutical Co. Ltd. (Seoul, Korea) according to the manufacturer’s instructions [29]. To determine the level of low-density lipoprotein (LDL) cholesterol, HDL level was subtracted from the total cholesterol level. Serum leptin and adiponectin levels were measured using ELISA kits (R&D Systems, Minneapolis, MN, USA).

### 2.8. RT-PCR, Bacterial DNA Extraction, and 16S rRNA Gene Sequencing

Total RNA was extracted from 3T3-L1 cells, mouse liver, and epididymal tissues using TRIzol (Invitrogen, Thermo Fisher) and the RNeasy Mini Kit (QIAGEN, Hilden, Germany). cDNA was synthesized and combined with specific primers and SYBR Green master mix (Bio-Rad, Hercules, CA, USA). The PCR primers used are listed in Table 1 (Macrogen, Seoul, Korea). Real-time RT-PCR was performed using the CFX connect real-time system (Bio-Rad, Hercules, CA, USA) [29]. Total genomic DNA was recovered from fecal samples using the PowerFecal Pro DNA Kit (QIAGEN, Hilden, Germany). The DNA concentration was measured using a Qubit 4 (Invitrogen, Thermo Fisher), and its purity was tested on 1% agarose gels. 16S rRNA sequencing was performed on two samples per group by combining DNA from five out of ten mice in equal proportions in each group (n = 2/group). The bacterial 16S rDNA gene was amplified using a specific V4 primer (forward primer, TCGTCGGCAGCGTCAGATGTGTATAAGAGCAGGTGCCAGCMGCC-GCGGTAA, and reverse primer, GTCTCGTGGGCTCGGAGATGTGTATAAGAGACA-GGGACTACHVGGGTWTCTAAT). Following purification, the samples were sequenced using an Illumina iSeq platform (San Diego, CA, USA) according to the manufacturer’s instructions. Trimmomatic (v0.39) was used to delete adaptor sequences from the 16S sequencing dataset. The data were grouped into operational taxonomic units (OTUs) and aligned using QIIME2 (version 1.9.5) (https://qiime2.org, accessed on 15 April 2022). α- and β-diversity analyses were performed using QIIME2 [32]. DNA extracted from the bacteria was amplified using selected bacterial primers (Macrogen, Seoul, Korea) and confirmed using qPCR (CFX System, Bio-Rad) to validate the results of 16S sequencing (n = 10 per group). The primer sequences used are listed in Table 2. The relative abundance of the selected bacterial species was computed as the ratio of total bacteria (F341/R518) [33].

### 2.9. Western Blotting

Proteins were extracted from 3T3-L1 cells and the liver using Pierce’s M-PER and T-PER protein extraction reagents (Rockford, IL, USA), to which a Halt protease inhibitor cocktail was added (Thermo Fisher). The homogenate was centrifuged at 14,000× *g* for 5 min at 4 °C, the supernatant was collected, and its concentration was quantified using the Bradford assay (Bio-Rad). SDS–PAGE gels were used to separate protein samples, which were subsequently transferred to polyvinylidene difluoride (PVDF) membranes. The membranes were blocked with 5% milk at room temperature for 1 h prior to overnight incubation at 4 °C with the primary antibodies UCP1, AMPK, p-AMPK, and β-actin (Cell Signaling Technology, Danvers, MA, USA). Next, secondary antibodies were added to the membranes for 1 h. Band intensity was determined using a chemiluminescence camera (Davinch K, Seoul, Korea) [34].

### 2.10. Statistical Analysis

Data were analyzed using Prism 8 (GraphPad Software, La Jolla, CA, USA). The results are presented as the mean ± SD. The *p*-values were calculated using one-way ANOVA (Dunnett’s test) or Student’s *t*-test, and the abundance of specific bacteria was determined using the non-parametric Kruskal–Wallis test. Statistical significance was set at *p* < 0.05.

## 3. Results

### 3.1. Identification of Bioactive Compounds in S. thunbergii Extract

The total phenolic content in the *S. thunbergii* extract was 4.46 ± 0.72 mg gallic acid equivalent/g extract, and the total flavonoid content was 28.16 ± 9.36 mg catechin equivalent/g extract. The bioactive substances in *S. thunbergii* extract were analyzed using HPLC and LC-MS. When the *S. thunbergii* extract and indole-2-carboxaldehyde, known as the primary ingredient, were subjected to HPLC, indole-2-carboxaldehyde was identified as one of the bioactive compounds in *S. thunbergii* extract with a retention time of 26.64 min and a concentration of 0.77 mg/g extract (Figure 1A). The LC-MS data also revealed the presence of indole-2-carboxaldehyde. In addition, the bioactive substances p-coumaric acid, cinnamic acid, l-rhamnulose, dulcitol, loliolide, and isovalerycarnitine were found in *S. thunbergii* extract (Figure 1B, Table 3). In addition, Vitamin D_3_ (C_27_H_44_O, *m*/*z* 385.3424, RT 2.165 min), Vitamin B_2_ (C_17_H_18_N_4_O_6_, *m*/*z* 359.1377, RT 3.329 min), and Vitamin B_2_ acid (C_17_H_18_N_4_O_7_, *m*/*z* 391.1216, RT 3.162 min) were detected.

### 3.2. Effects of S. thunbergii Extract and Indole-2-Carboxaldehyde on the Adipogenic Differentiation of 3T3-L1 Pre-Adipocytes

To determine the cytotoxicity of *S. thunbergii* (ST) extract and the index component indole-2-carboxaldehyde (I2C), a cell viability test was performed in 3T3-L1 preadipocytes. As shown in Figure 2A, ST and I2C showed more than 80% cell viability at concentrations below 20 mg/L. Therefore, a continuous experiment was conducted at a concentration of 20 mg/L. To evaluate anti-adipogenic effects, 3T3-L1 preadipocytes were cultured for 6 days in adipocyte differentiation media with or without ST and I2C. As shown in Figure 2B,C, ST had a dose-dependent effect on lowering Oil Red O staining to 80.46% and 76.79% at 10 and 20 mg/L, respectively, compared with the PBS-treated control. I2C also reduced lipid droplet accumulation by 81.52% and 85.75% at 10 and 20 mg/L, respectively, in 3T3-L1 adipocytes compared to that in the PBS-treated control.

### 3.3. Effects of S. thunbergii Extract on the Expression of Adipogenic and Thermogenic Genes and Proteins in Differentiated 3T3-L1 Cells

To determine the anti-adipogenic mechanism of ST extract, we analyzed expression levels of genes associated with adipogenesis, lipogenesis, and thermogenesis. ST treatment significantly reduced the expression of adipogenesis-related genes *C/ebpα* (by 54%) and *Pparγ* (by 72%) at 20 mg/L compared to that in the PBS-treated controls (Figure 3A,B). I2C treatment significantly decreased *Srebp1* gene expression by 44% at 20 mg/L; however, ST had no effect on *Srebp1* (Figure 3C). In addition, compared to the PBS-treated control, ST treatment significantly elevated expression of genes associated with energy sensing and adipocyte browning, including *Ampk* (by 225%), *Sirt1* (by 790%), *Pgc-1a* (by 121%), and *Ucp1* (by 608%) at 20 mg/L, whereas I2C treatment only increased *Ampk* (by 247%) and *Sirt1* (by 223%) at 20 mg/L (Figure 3D–H). ST and I2C did not significantly change the expression of Pparα (Figure 3G). Lipogenesis-related genes, such as *Acc* and *Fas* were also measured, but no statistically significant alterations were observed (data not shown). In addition, ST treatment increased the ratio of pAMPK/AMPK and UCP1 protein expression (Figure 3I).

### 3.4. Effects of S. thunbergii Extract on Body and Organ Weights in High Fat-Fed C57BL/6 Mice

When ST extract was administered to animals for 8 weeks (Figure 4A), changes in body weight were detected. Body weight in the ST low- and high-concentration groups was significantly lower from the seventh week onwards, as shown in Figure 4B, compared to the HF group. The HF diet significantly increased the final body weight gain, while ST treatment reduced it compared to that in the Con group (Figure 4C). Figure 4D shows no difference in food intake between the groups with high-fat consumption. Compared with the HF group, all ST groups had significantly lower epididymal fat weights (Figure 4E). The liver weight was also measured, and there was no difference between the groups (data not shown). Compared to the HF group, ST treatment was more effective in reducing body weight and fat.

### 3.5. Effects of S. thunbergii Extract on Histology of Liver and Adipose Tissue in High-Fat-Fed C57BL/6 Mice

The size of the adipocytes in the fat tissue and the histology of the liver were investigated using H&E staining. As shown in Figure 5A,B, the area of adipocytes in the epididymal white adipose tissue was 2.2 times greater in the HF group than in the Con group, whereas the ST treatment groups (150 and 300 mg/kg bw) had significantly reduced adipocyte size (36% and 53%, respectively) compared to the HF group. Hepatocellular ballooning is an indicator of an inflamed fatty liver caused by obesity or alcohol consumption [35]. In the liver, the HF group had a greater number of ballooned hepatocytes than the Con group, but the ST treatment group had fewer ballooned hepatocytes than the HF group (Figure 5C). Lipid accumulation in the liver was investigated using Oil Red O staining; however, no substantial lipid droplet production was detected in the liver (data not shown).

### 3.6. Effects of S. thunbergii Extract on the Serum Profiles in High-Fat-Fed C57BL/6 Mice

The effects of ST extract on the prevention of hyperlipidemia and hyperglycemia were investigated. Compared to the Con group, the HF diet considerably increased the levels of all serum profiles, as indicated in Figure 6A–D. However, when compared to the HF group, the high ST concentration group (300 mg/kg bw) only lowered LDL cholesterol by 28%. Leptin and adiponectin are obesity-related hormones. As demonstrated in Figure 6E,F, there was no significant change in adiponectin levels, but both ST groups had lower leptin levels than the HF group by 39% and 44%, respectively. Moreover, the HF diet increased fasting glucose levels compared to those in the Con group, but both ST treatments reduced fasting glucose levels (Figure 6G). The HF group showed increased AUC of the glucose tolerance test (OGTT), whereas the ST high concentration group (300 mg/kg bw) showed lowered glucose tolerance (Figure 6I). In comparison with the HF group, we found that ST extract has the potential to prevent hyperlipidemia and hyperglycemia.

### 3.7. Effects of S. thunbergii Extract on AMPK Activation in White Adipose Tissue and Liver in High-Fat-Fed C57BL/6 Mice

To elucidate the anti-obesity metabolic mechanism of ST extract in animal experiments, we measured the expression levels of genes associated with adipogenesis, lipogenesis, and adipocyte browning. As a result, as shown in Figure 7A–D, the expression of *Ampk* and *Sirt1* increased in fat (121% and 241%, respectively) and liver (1044% and 252%, respectively) in the ST high-concentration group compared to the HF group. The other genes that were altered in the in vitro experiment did not change in vivo (data not shown). Furthermore, when compared to the HF group, both ST treatments increased the pAMPK/AMPK ratio (Figure 7E,F). It has been shown that ST extract can reduce obesity in mice fed a high-fat diet by activating AMPK.

### 3.8. Effects of S. thunbergii Extract on the Gut Microbiota Composition in High-Fat-Fed C57BL/6 Mice

Dietary modifications, particularly high-fat diets, can alter the composition of gut microbiota [31]. Additionally, we examined the effects of a high dose of ST extract (300 mg/kg body weight) on the fecal microbiota of obese mice. As seen in Figure 8A, the richness and evenness of the gut microbiota were assessed. The Shannon diversity index (richness) was significantly lower in the HF group than in the Con group and somewhat higher in the ST group than in the HF group, although this difference was not statistically significant (*p* = 0.29). The same tendency was also observed in the ASVs index (evenness), although the differences were not statistically significant (Con vs. HF; *p* = 0.22, HF vs. HF-StH; *p* = 0.35; Figure 8A). The similarity of bacterial distribution was shown to differ between the Con and HF groups (R = 1, *p* = 0.326) in the β analysis based on the weighted UniFrac distance, but not between the HF and HF-StH groups (R = 0, *p* = 0.657; Figure 8B). As shown in Figure 8C, according to the bacterial classification, when searching by phylum, a high-fat diet increased the relative abundance of Firmicutes (green color) compared to the Con group, while the abundance of Bacteroidetes (yellow color) decreased. ST treatment did not significantly improve the condition. The Firmicutes/Bacteroidota ratio, an indicator of obesity, was significantly higher in the high-fat group compared to the Con group, although it was on the decline in the ST group compared to the HF group (*p* = 0.57; Figure 8D). When we screened specific genera and species whose relative abundance was modified by high-fat diet or treatment (*p* < 0.2), the abundance of *Lactobacillus* spp. was reduced (*p* = 0.19) and *Lactococcus* spp. increased in response to a high-fat diet; ST administration did not reverse these changes (Figure 8E). Similar results were obtained when we used qPCR to measure the species levels of *Lactobacillus reuteri* and *Lactococcus lactis* associated with the genus (Figure 8F). The abundance of *Bacteroides vulgatus* was less common and *Romboutsia ilealis* was more prevalent (*p* = 0.18) at the species level in response to the high-fat diet, although ST treatment tended to reverse these trends (Figure 8C,E). Similar results were obtained by qPCR (Figure 8F). When we used qPCR to detect the levels of probiotics known to be affected by phytochemicals [36], the levels of *Faecalibacterium prausnitzii* decreased in the HF group compared to the Con group and increased in the high-concentration ST group (Figure 8F). When we calculated the relationship between the qPCR results and the obesity-related indices, *Lactococcus lactis* had negative correlations with some obesity-related indices, whereas *Bacteroides vulgatus* had positive correlations with the obesity-related markers (Figure 8G). Consequently, it is indicated that the ST extract may enhance the gut microbiota composition to improve obesity.

## 4. Discussion

In this study, we investigated the anti-obesity properties of ethanol extract from *Sargassum thunbergii*, which is rich in indole-2-carboxyaldehyde, on 3T3-L1 adipocytes and in an obese mouse model. Our research demonstrated that *S**. thunbergii* extract exerted anti-obesity effects in vitro and in vivo through AMPK activation and alteration of gut microbiota composition, contrary to our hypothesis that *S. thunbergii* extract may attenuate obesity by lowering adipogenesis and lipogenesis, inducing browning, and modulating gut microbiota composition. Brown algae, called *S. thunbergii*, are known to contain bioactive compounds such as polyphenols (gallic acid, pyrogallol, tannins), terpenoids, alkaloids (derivatives of indoles), and polysaccharides such as fucoidan [25,37], which have been linked to antioxidant, anti-inflammatory, and anti-obesity benefits [22,23,24]. However, the underlying mechanisms of the effect of *S. thunbergii* on obesity and gut microbiota composition are not fully understood.

In an in vitro assay, following HPLC confirmation of indole-2-carboxyaldehyde as a component of *S. thunbergii*, we treated cells with both *S. thunbergii* extract and an indole-2-carboxyaldehyde standard during adipocyte differentiation and confirmed that the number of fat droplets decreased in a dose-dependent manner in mature adipocytes. *S. thunbergii* extract exerted anti-obesity effects by suppressing the expression levels of genes related to adipogenesis [38] and increasing the expression levels of genes and proteins related to thermogenesis [39] by boosting AMPK compared to the PBS control. In a previous study, indole-2-carboxaldehyde derived from *S. thunbergii* reduced adipogenesis by inhibiting *Srebp-1c*, *Pparγ*, and *C/ebpα* while activating *Ampk* in 3T3-L1 cells [25]. However, our data indicated that indole-2-carboxyaldehyde only reduced *Srebp1* gene expression while increasing *Sirt1* and *Ampk* gene expression levels compared to those in the PBS control. Therefore, it is considered that the anti-obesity benefits of *S. thunbergii* extract are a result of the interaction of multiple natural chemicals in *S. thunbergii*, not simply indole-2-carboxyaldehyde. Taken together, the anti-obesity effects of *S. thunbergii* extract may involve the inhibition of adipogenesis and the stimulation of thermogenesis to convert white fat to beige fat to generate heat in 3T3-L1 cells.

Hyperglycemia and hyperlipidemia are potential targets for the prevention and treatment of obesity [40]. In our animal investigations, a high-fat diet significantly induced body weight and fat mass, and induced hyperglycemia and hyperlipidemia through the induction of TG, T-chol, HDL-C, and LDL-C levels, as well as fasting glucose and oral glucose tolerance levels. In comparison to animals administered an HF diet, *S. thunbergii* extract administration decreased body weight, fat mass, LDL-cholesterol, fasting blood glucose levels, and oral glucose tolerance. Leptin is an adipokine produced by adipocytes that plays a role in regulating food intake and energy expenditure. Hyperleptinemia is associated with cardiovascular risk [41]. Leptin levels were increased with the high-fat diet, and ST treatment markedly decreased leptin levels, which may have positive effects on cardiovascular risk. This suggests that *S. thunbergii* extract can prevent hyperglycemia and hyperlipidemia in obese mice.

AMPK is a master regulator of energy balance that controls the metabolism of glucose, lipids, and mitochondria [42]. AMPK has been highlighted as a target for obesity prevention and therapy because of its role in suppressing fatty acid synthesis and inducing β-oxidation by inhibiting ACC and SREBP1 [43], as well as in the development of mitochondrial biogenesis and adipocyte browning by activating UCP1 [9]. Sirt1 is also involved in energy balance by regulating glucose homeostasis and fat metabolism, similar to AMPK, making it an additional target for the development of therapies for metabolic diseases caused by obesity [44]. Numerous natural compounds have been shown to promote AMPK activation, resulting in various health benefits [45]. In a previous study, *S. thunbergii* alleviated fatty liver and reduced fat accumulation in obese mice by inhibiting *Pparγ* and increasing the expression of *Ucp-1* and *Ucp-3* genes [26]. Nevertheless, according to our findings, *S. thunbergii* treatment reduced body weight, fat weight, and adipocyte size in fat tissues compared to the HF group by increasing *Ampk* and *Sirt1* gene expression in the WAT and liver. Thus, *S. thunbergii* extract can prevent hyperlipidemia and obesity by activating AMPK.

Recently, studies have been conducted on the role of gut microbiota in obesity and the role of natural compounds in preventing obesity by altering the gut microbiota composition [46]. We, too, examined the effect of *S. thunbergii* on the fecal microbiota of HFD-induced obese mice. In response to a high-fat diet, bacterial diversity was reduced considerably, and bacterial distribution was altered compared to the control. The ratio of Firmicutes to Bacteroidetes indicates the instability of the gut microbiota in obese individuals [47], and this ratio is also altered by a high-fat diet. *S. thunbergii* extract resulted in a minor improvement in the gut microbiota alterations generated by a high-fat diet, particularly in bacterial alterations at the species level. *Bacteroides vulgatus* is known to play an important role in enhancing brown fat metabolism and protecting against obesity [48]. *Faecalibacterium prausnitzii* is known to contribute to gut homeostasis by producing the anti-inflammatory metabolite butyrate [49]. *Romboutsia ilealis* has been identified as an indicator of gut dysbiosis [50]. In our study, in response to the high-fat diet, *Bacteroides vulgatus* and *Faecalibacterium prausnitzii* decreased, but *Romboutsia ilealis* increased. As a result of *S. thunbergii* extract treatment, however, the alteration appears to have improved. This indicates that the *S. thunbergii* extract may have the potential to improve the amount of probiotics and gut dysbiosis, in turn improving obesity.

In a recent study, 100 and 300 mg/kg bw of *S. thunbergii* extract in HFD-induced obese mice reduced body weight and fat accumulation without any toxicity or deleterious effects [26]. Consequently, a similar dose was chosen for this investigation. Using the body surface area normalization method [51,52], the human equivalent amount of *S. thunbergii* extract was determined, and a dose of 25 mg/kg/day was selected for humans. Therefore, a person weighing 70 kg would need to consume approximately 1750 mg of *S. thunbergii* extract daily to achieve comparable benefits. The limitation of this experiment is the short duration of the animal study (8 weeks). Based on the results of previous experiments and our in vitro experiments, we hypothesized that *S. thunbergii* treatment would improve the expression of genes involved in adipogenesis and thermogenesis in an animal model; instead, only energy sensors, *Ampk* and *Sirt1* levels, were induced. In addition, there was no difference in liver weight across the groups, and no fat accumulation was observed in the liver. Previous studies have reported that hepatic steatosis was induced after 12 weeks of feeding a HF diet to mice [53]; therefore, we halted the experiment before the adipogenesis stage. To evaluate the effects of phytochemicals on adipogenesis in a mouse model, it is necessary to feed the mice a HF diet for more than 12 weeks. Another constraint was that the number of samples for 16S sequencing was insufficient, making it difficult to obtain significant results; hence, qPCR was performed using individual DNA from each mouse feces to augment the data, and the conclusions of the study were made based on the qPCR results rather than the sequencing results. In addition, additional experiments are required to investigate how this modification in gut microbiota composition by *S. thunbergii* extract affects the production of microbe-derived factors such as short chain fatty acids.

## 5. Conclusions

This study demonstrated that *S. thunbergii* extract significantly suppressed fat accumulation via suppression of adipogenic genes and induction of thermogenic genes in vitro; decreased body weight gain, white adipose tissue weight, low-density lipoprotein-cholesterol, and serum leptin levels; and improved glucose intolerance via AMPK activation in high-fat-fed mice. In addition, *Bacteroides vulgatus* and *Faecalibacterium prausnitzii*, which are probiotics, increased in response to *S. thunbergii* extract administration, although *Romboutsia ilealis,* which is associated with gut dysbiosis, decreased, thereby improving the imbalance in the intestinal microflora. Based on these improvements, *S. thunbergii* extract could prevent obesity in mice raised to develop obesity through a high-fat diet via modulating AMPK activation and gut microbiota composition. The findings of this study suggest that *S. thunbergii* may be utilized as an anti-obesity agent.

## Figures and Tables

**Figure 1 foods-11-02529-f001:**
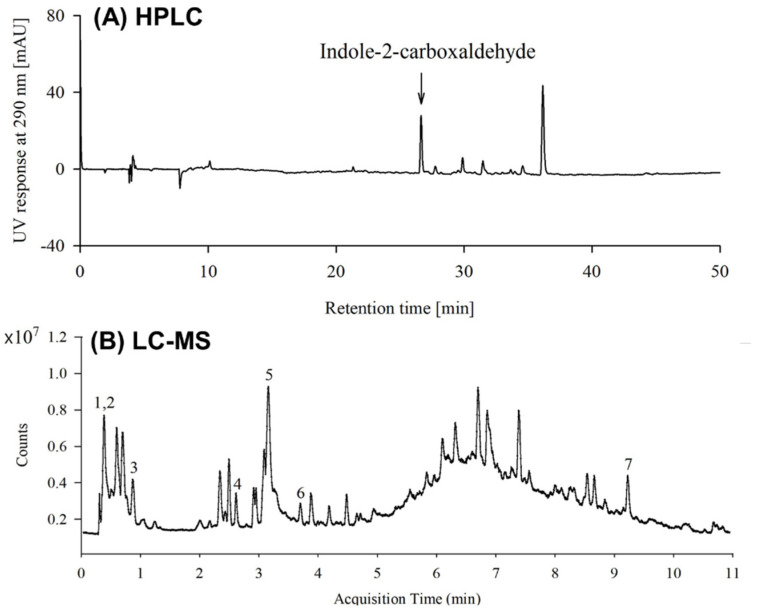
Representative (**A**) HPLC (290 nm) and (**B**) LC-MS chromatogram of *S*. *thunbergii* extract. 1. L-rhamnulose, 2. Dulcitol, 3. p-Coumaric acid, 4. Indole-2-carboxaldehyde, 5. Isovalerycanitine, 6. Loliolide, and 7. Cinnamic acid.

**Figure 2 foods-11-02529-f002:**
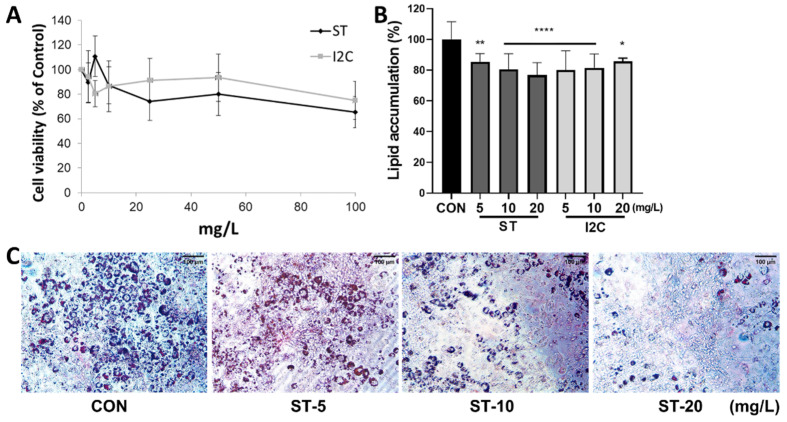
Effects of *S. thunbergii* (ST) extract and indole-2-carboxaldehyde (I2C) on cell viability and Oil Red O staining in 3T3-L1 cells. (**A**) Pre-adipocytes were treated with 0–100 mg/L of *S. thunbergii* and indole-2-carboxaldehyde for 48 h, and viability was determined using the MTS assay. (**B**) 3T3-L1 cells were induced to differentiation in the presence of *S. thunbergii* extract and indole-2-carboxaldehyde (0–20 mg/L) for 10 days. Extracted Oil Red O was quantified at OD 500 nm. (**C**) Representative images of Oil Red O staining. All data are presented as means ± SD (n = 8). Compared with the CON treatment, * *p* < 0.05, ** *p* < 0.01, **** *p* < 0.0001 by one way ANOVA followed by Dunnett’s test.

**Figure 3 foods-11-02529-f003:**
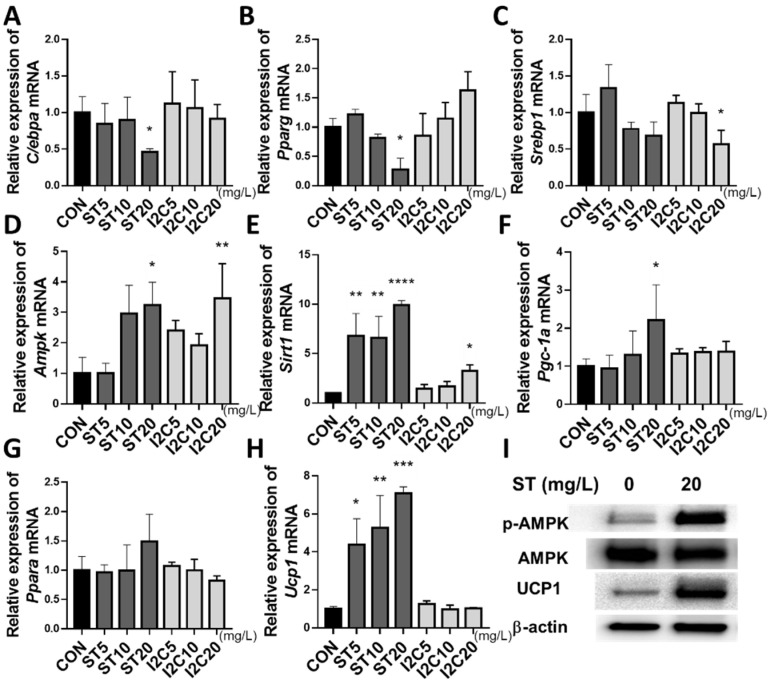
Effects of *S. thunbergii* (ST) extract and indole-2-carboxaldehyde (I2C) on the adipogenesis- and thermogenesis-related gene and protein expressions in differentiated 3T3-L1 cells. (**A**) *C/ebpa*, (**B**) *Pparg*, (**C**) *Srebp1*, (**D**) *Ampk*, (**E**) *Sirt1*, (**F**) *Pgc-1α*, (**G**) *Pparα*, (**H**) *Ucp1* mRNA. (**I**) Western blot. 3T3-L1 cells were induced to differentiation in the presence of *S. thunbergii* extract and indole-2-carboxaldehyde (0–20 mg/L) for 10 days, and RNA and protein were isolated. β-actin was amplified under the same PCR conditions for normalized quantitative data and also used as a loading control in Western blot. Results are expressed as means ± SD (n = 3). Compared with the CON treatment, * *p* < 0.05, ** *p* < 0.01, *** *p* < 0.001, **** *p* < 0.0001 by one way ANOVA followed by Dunnett’s test.

**Figure 4 foods-11-02529-f004:**
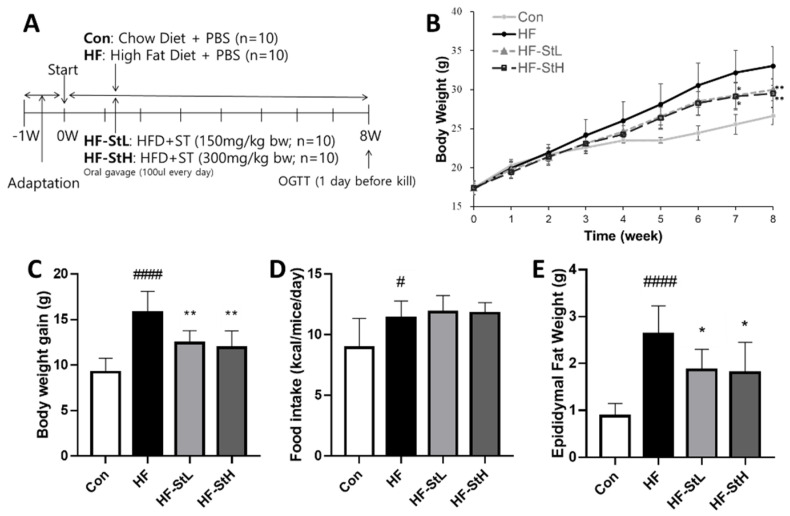
Effects of *S. thunbergii* (ST) extract on body weight and organ weight in high-fat-fed C57BL/6 mice. (**A**) Experimental timeline. (**B**) Trend of body weight change for 10 weeks. (**C**) Body weight gain. (**D**) Average food intake for 10 weeks. (**E**) Epididymal fat weight. Con: chow diet, HF: high-fat diet, HF-StL: high-fat diet+ST extract 150 mg/kg bw, HF-StH: high-fat diet+ST extract 300 mg/kg bw. All data are presented as means ± SD (n = 10). Compared between the control and the high-fat diet group; # *p* < 0.05, #### *p* < 0.0001 by student *t*-test. Compared with the high-fat diet group, * *p* < 0.05, ** *p* < 0.01 by one way ANOVA followed by Dunnett’s test.

**Figure 5 foods-11-02529-f005:**
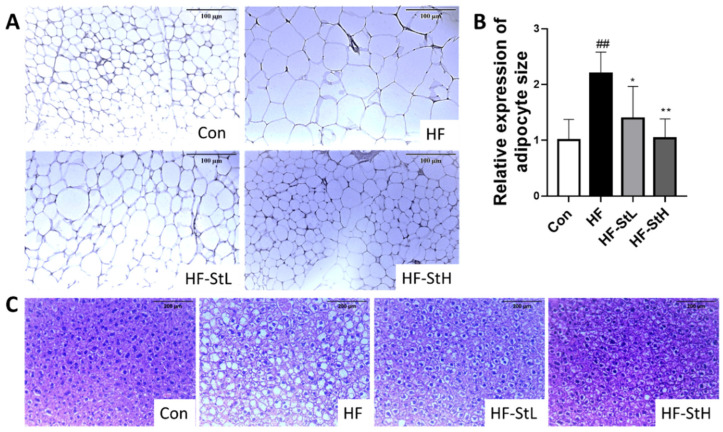
Histological changes in white adipose tissue (WAT) and liver in high-fat-fed C57BL/6 mice. (**A**) White adipose tissue section by H&E staining (×100). (**B**) Average adipocyte size. (**C**) Liver section by H&E staining (×200). Con: chow diet, HF: high-fat diet, HF-StL: high-fat diet+ST extract 150 mg/kg bw, HF-StH: high-fat diet+ST extract 300 mg/kg bw. All data are presented as means ± SD (n = 10). Compared between the control and the high-fat diet group, ## *p* < 0.01 by student *t*-test. Compared with the high-fat diet group, * *p* < 0.05, ** *p* < 0.01 by one way ANOVA followed by Dunnett’s test.

**Figure 6 foods-11-02529-f006:**
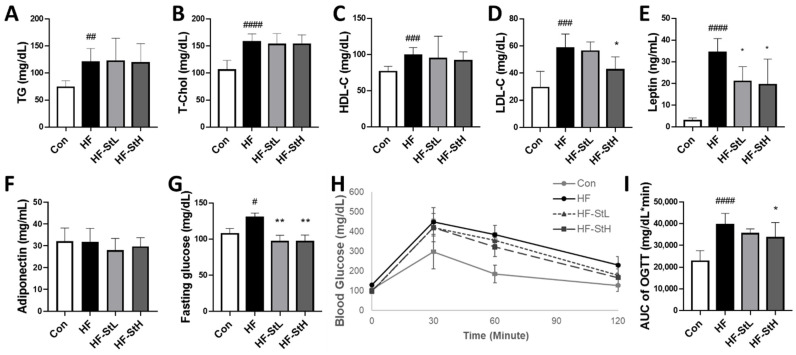
Effects of *S. thunbergii* (ST) extract on the serum profiles and the glucose levels in high fat-fed C57BL/6 mice. (**A**) Triacylglycerol. (**B**) Total cholesterol. (**C**) HDL-cholesterol. (**D**) LDL-cholesterol. (**E**) Serum leptin. (**F**) Serum adiponectin levels. (**G**) Fasting blood glucose level. (**H**) Curve of OGTT. (**I**) AUC of OGTT. Con: chow diet, HF: high-fat diet, HF-StL: high-fat diet+ST extract 150 mg/kg bw, HF-StH: high-fat diet+ST extract 300 mg/kg bw. All data are presented as means ± SD (n = 10). Compared between the control and the high-fat diet group, # *p* < 0.05, ## *p* < 0.01, ### *p* < 0.001, #### *p* < 0.0001 by Student’s *t*-test. Compared with the high-fat diet group, * *p* < 0.05, ** *p* < 0.01 by one way ANOVA followed by Dunnett’s test.

**Figure 7 foods-11-02529-f007:**
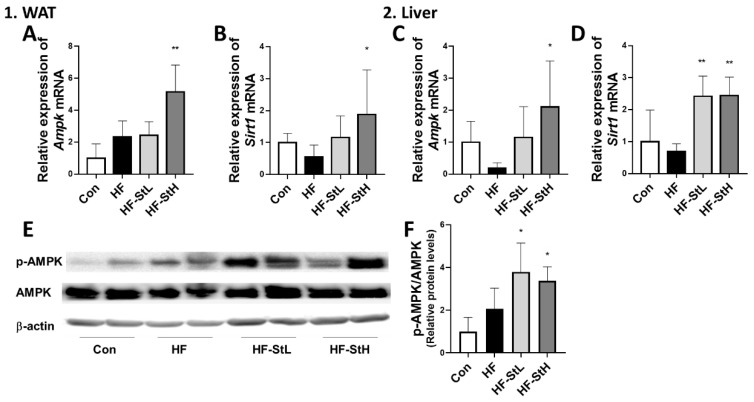
Effects of *S. thunbergii* (ST) extract on *Ampk/Sirt1* pathway in white adipose tissue (WAT) and liver. (**A**) *Ampk* and (**B**) *Sirt1* mRNA expressions in WAT. (**C**) *Ampk* and (**D**) *Sirt1* mRNA expressions in liver. (**E**) Protein expressions of p-AMPK, AMPK, and β-actin in liver tissue. (**F**) Ratios of p-AMPK/AMPK. Con: chow diet, HF: high-fat diet, HF-StL: high-fat diet+ST extract 150 mg/kg bw, HF-StH: high-fat diet+ST extract 300 mg/kg bw. All data are presented as means ± SD (n = 10). Compared with the high-fat diet group, * *p* < 0.05, ** *p* < 0.01 by one way ANOVA followed by Dunnett’s test.

**Figure 8 foods-11-02529-f008:**
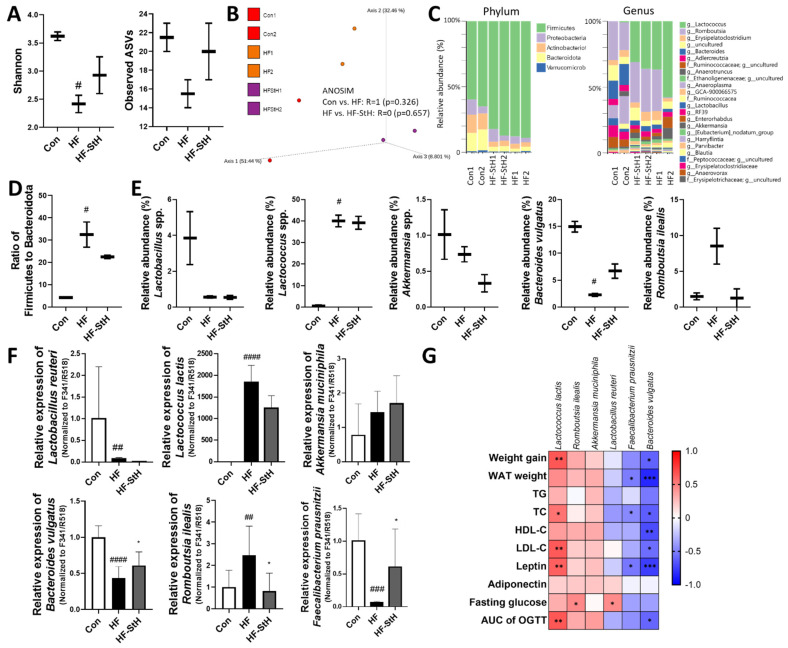
Effects of *S. thunbergii* (ST) extract on the gut microbiota composition in high-fat-fed C57BL/6 mice. (**A**) Shannon evenness and observed ASVs richness diversity. (**B**) Principal coordinate analysis (PCoA) of weighted UniFrac distances. (**C**) Taxonomy community analysis at phylum and genus level. (**D**) The ratio of Firmicutes to Bacteroidota. (**E**) Specific taxa whose relative abundance was changed by high-fat diet or treatment (n = 2; fecal DNA from 5 mice were pooled into a sample for 16S sequencing). (**F**) Quantitative PCR results for selected bacterial groups. The relative abundance of bacterial groups was expressed as a ratio of total bacteria (F341/R518). Data are expressed as means ± SD (n = 10/group). Con: chow diet, HF: high-fat diet, HF-StL: high-fat diet+ST extract 150 mg/kg bw, HF-StH: high-fat diet+ST extract 300 mg/kg bw. Compared between the control and the high-fat diet group; # *p* < 0.05, ## *p* < 0.01, #### *p* < 0.0001 by non-parametric Kruskal–Wallis test. Compared between the high-fat diet and the experimental group, ^⁎^
*p* < 0.05 by non-parametric Kruskal–Wallis test. (**G**) Correlation analysis of gut microbial species and obesity-related markers (Spearman correlation; * *p* < 0.05, ** *p* < 0.01, *** *p* < 0.001).

**Table 1 foods-11-02529-t001:** Sequences of primers used for PCR.

Gene	Forward Primer	Reverse Primer
*β* *-actin*	CCCTACAGTGCTGTGGGTTT	GAGACATGCAAGGAGTGCAA
*Ampk*	TGTTCCAGCAGATCCTTTCC	ATAATTGGGTGAGCCACAGC
*C/ebpα*	ATCAGCGCCTACATTGATCC	TTGCTTGGCTGTCGTAGATG
*Pparγ*	CCCTGGCAAAGCATTTGTAT	GAAACTGGCACCCTTGAAAA
*Pgc1α*	AATGCAGCGGTCTTAGCACT	GTGTGAGGAGGGTCATCGTT
*Pparα*	TCTTCACGATGCTGTCCTCCT	CTATGTTTAGAAGGCCAGGC
*Sirt1*	AGTTCCAGCCGTCTCTGTGT	CTCCACGAACAGCTTCACAA
*Srebp-1*	GAGCCTTCAGACACGTCCTC	ACTCTTCTGGTGTGGCTGCT
*Ucp1*	CTGCCAGGACAGTACCCAAG	GCCACAAACCCTTTGAAAAA

**Table 2 foods-11-02529-t002:** Sequences of primers used for bacterial profiling.

Target	Forward Primer	Reverse Primer	Reference
Uni (F341/R518)	CCTACGGGAGGCAGCAGT	ATTACCGCGGCTGCTGG	Lubbs (2009)
*Akkermansia muciniphila*	CTGAACCAGCCAAGTAGCG	CCGCAAACTTTCACAACTGACTTA	Collado (2007)
*Bacteroides vulgatus*	GCATCATGAGTCCGCATGTTC	TCCATACCCGACTTTATTCCTT	Wang (1996)
*Faecalibacterium prausnitzii*	AGATGGCCTCGCGTCCGA	CCGAAGACCTTCTTCCTCC	Wang (1996)
*Lactobacillus reuteri*	GCCGCCTAAGGTGGGACAGAT	AACACTCAAGGATTGTCTGA	Walter (2000)
*Lactococcus lactis*	TGAAGAATTGATGGAACTCG	CATTGTGGTTCACCGTTC	Bachmann (2015)
*Romboutsia ilealis*	GGGGCTAGCGTTATTCCGAA	CACCTGTCACTTCTGTCCCC	Designed in this study

**Table 3 foods-11-02529-t003:** Chemical characterization of *S. thunbergii* extract by LC-MS.

Peak	Compound Name	Molecular Formula	Molecular Weight	Retention Time (min)	Calcd. *m*/*z* [M + H]^+^	Content *
1	L-rhamnulose	C_6_H_12_O_5_	164.16	0.391	165.0546	1.53
2	Dulcitol	C_6_H_14_O_6_	182.17	0.391	183.0863	1.53
3	p-Coumaric acid	C_9_H_18_O_3_	164.16	0.857	165.0757	0.64
4	Indole-2-carboxaldehyde	C_9_H_7_NO	145.16	2.614	146.0600	0.77
5	Isovalerycanitine	C_12_H_23_NO_4_	245.31	3.085	246.1700	0.85
6	Loliolide	C_11_H_16_O_3_	196.24	3.689	197.1172	0.49
7	Cinnamic acid	C_9_H_8_O_2_	148.16	9.232	149.0597	0.86

* ug Indole-2-carboxaldehyde/mg extract.

## Data Availability

The date are available from the corresponding author.

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
