# Peer review of "Sargassum thunbergii Extract Attenuates High-Fat Diet-Induced Obesity in Mice by Modulating AMPK Activation and the Gut Microbiota"

_foods, 2022, doi:10.3390/foods11162529_

Round 1

Reviewer 1 Report

This study investigated the effect of Sargassum thunbergii 's ethanol extract on weight loss using cell and animal experiments and explored the mechanism of action. Train of thought is clearer, logic is better! However, there are also some problems in the manuscript, for example, whether it is the extract used or Sargassum thunbergii 's explanation is not clear, which is confusing; The number of intestinal flora determined is too low, the specific composition of the mixture (a 70% ethanol extract of S. thunbergii) is not determined, etc., and some other problems are described below.

1. In the manuscript, 70% ethanol extract of Sargassum thunbergii (not Sargassum thunbergii) reduced obesity in high-fat diet-induced mice by modulating AMPK activation and gut microbiota, so the title should be changed to reflect the content of the study. The abstract is about research using S. thunbergii, and lines 72-73 are about research using S. thunbergii extract ! The current research is very unclear on these two small statements, please explain clearly in the author's manuscript.

2. Due to the mixture used in the experiment (S. thunbergii extract extracted with 70% ethanol), the composition needs to be analyzed to provide nutrients and active ingredients, such as phenols, flavonoids and so on.

3. Indole-2-carboxaldehyde is the main active ingredient extracted from S. thunbergii Why is there no quantitative analysis in the manuscript? Please provide the content of active ingredient that can be detected at present (especially the seven active ingredients shown in Figure 1)?

4. It is not clear in the notes that * stands for comparison with which group(e.g., Figure 2 and Figure 3).

5. The H&E staining in Figure 5 should add a ruler to the figure!

6. There were 10 mice in each group, why only 5 mice were tested for intestinal flora?

The five mouse assays were actually mixed and only two were assayed, requiring at least five mice to be assayed individually. The test of 5 mice is a little small, generally more than 8 mice are needed.

7. 16S rRNA gene sequencing is generally recognized as only the genus level. In Fig. 8C, the author mentioned the species water, please explain the reason. In addition figure 8C needs to indicate which species each color represents.

8. Some indicators of obesity need to be associated with the flora at the genus level or species level.

9. Why did the authors not consider the determination of metabolites of intestinal flora such as SCFAs in faeces?

Author Response

  1. In the manuscript, 70% ethanol extract of Sargassum thunbergii (not Sargassum thunbergii) reduced obesity in high-fat diet-induced mice by modulating AMPK activation and gut microbiota, so the title should be changed to reflect the content of the study. The abstract is about research using S. thunbergii, and lines 72-73 are about research using S. thunbergii extract ! The current research is very unclear on these two small statements, please explain clearly in the author's manuscript.

Thank you for the advice. We changed the title, and have now better clarified that we used extract throughout.

  1. Due to the mixture used in the experiment (S. thunbergii extract extracted with 70% ethanol), the composition needs to be analyzed to provide nutrients and active ingredients, such as phenols, flavonoids and so on.

We have added data of total phenolic and flavonoid contents in the extract (Line 203-205).

  1. Indole-2-carboxaldehyde is the main active ingredient extracted from S. thunbergii Why is there no quantitative analysis in the manuscript? Please provide the content of active ingredient that can be detected at present (especially the seven active ingredients shown in Figure 1)?

We have added the quantitative analysis of the seven ingredients in the extract to Table 3 (Line 218).

  1. It is not clear in the notes that * stands for comparison with which group(e.g., Figure 2 and Figure 3).

We have indicated that the asterisk represents comparisons with the control group in the legends for Figure 2 and 3 (Line 237, 274).

  1. The H&E staining in Figure 5 should add a ruler to the figure!

We have added a ruler to the top right corner of images in Figure 5.

  1. There were 10 mice in each group, why only 5 mice were tested for intestinal flora? The five mouse assays were actually mixed and only two were assayed, requiring at least five mice to be assayed individually. The test of 5 mice is a little small, generally more than 8 mice are needed.

Due to cost, 16S rRNA sequencing was performed on two samples per group by combining the DNA from five out of ten mice in equal proportions in each group. We modified the sentence (Line 163-165). In addition, due to the small number in the sequencing, we additionally performed qPCR with DNA from each mouse. It was mentioned in the discussion section as the study limitation (Line 485-487).

  1. 16S rRNA gene sequencing is generally recognized as only the genus level. In Fig. 8C, the author mentioned the species water, please explain the reason. In addition, figure 8C needs to indicate which species each color represents.

We have replaced Fig 8C with a genus-level image and added the color-representation indicator.

  1. Some indicators of obesity need to be associated with the flora at the genus level or species level.

Figure 8G has been updated to include a heatmap depicting the relationship between obesity-related indices and gut microbiota at the species level. For the comparison, we used qPCR data instead of 16S-sequencing data.

  1. Why did the authors not consider the determination of metabolites of intestinal flora such as SCFAs in faeces?

Since our laboratory has not yet prepared the setting for measuring SCFAs, we intend to conduct additional experiments to measure the amount of microbial-derived factors once the environment has been prepared. We have stated it as the study limitation in the discussion (Line 487-490).

Reviewer 2 Report

The manuscript by Dahee Kim et al. investigated the effect of extracts from Sargassum thunbergii on HFD-induced obesity in mice. The study is of interest to the readers and I have the following suggestions:

1, for the principal coordinate analysis in figure 8, each dot should represent one mouse. 

2, for the comparison of the gut microbiota, a Lefse analysis should be used. 

3, Is it possible to analyze the SCFAs levels in the feces? This would help to illustrate the beneficial effects of Sargassum thunbergii on gut microbiota. 

Author Response

1, for the principal coordinate analysis in figure 8, each dot should represent one mouse.

Due to cost, 16S rRNA sequencing was performed on two samples per group by combining the DNA from five out of ten mice in equal proportions in each group. So each dot represents one sample used for sequencing. Due to the small sample size in the sequencing, we additionally performed qPCR with DNA from each mouse. It was mentioned in the discussion section as the study limitation (Line 485-487).

2, for the comparison of the gut microbiota, a Lefse analysis should be used.

We ran Lefse analysis but found no significant results, possibly due to the small number of samples.

3, Is it possible to analyze the SCFAs levels in the feces? This would help to illustrate the beneficial effects of Sargassum thunbergii on gut microbiota.

Since our laboratory has not yet prepared the setting for measuring SCFAs, we intend to conduct additional experiments to measure the amount of microbial-derived factors once the environment has been prepared. We have stated it as the study limitation in the discussion (Line 487-490).

Round 2

Reviewer 1 Report

1. Should the manuscript be in tracked revision mode or marked in red?

2. Due to the mixture used in the experiment (S. thunbergii extract extracted with 70% ethanol), the composition needs to be analyzed to provide nutrients and active ingredients, such as phenols, flavonoids and so on.

Too few ingredients are provided.

3. Indole-2-carboxaldehyde is the main active ingredient extracted from S. thunbergii Why is there no quantitative analysis in the manuscript? Please provide the content of active ingredient that can be detected at present (especially the seven active ingredients shown in Figure 1)?

What are the units of this content? Calculate the absolute quantity

4.  The H&E staining in Figure 2 should add a ruler to the figure!

Author Response

  1. Should the manuscript be in tracked revision mode or marked in red?

We apologize for not attaching the manuscript with tracked revision mode the previous time. The attached manuscript “changes highlighted 1” is the previous version, and the file “changes highlighted 2” is the current version.

  1. Due to the mixture used in the experiment (S. thunbergii extract extracted with 70% ethanol), the composition needs to be analyzed to provide nutrients and active ingredients, such as phenols, flavonoids and so on.

- Too few ingredients are provided.

We screened the composition of the ingredients in the LC results again, but only found a few more vitamins. We included the names of vitamins in the results (Line 212-214).

  1. Indole-2-carboxaldehyde is the main active ingredient extracted from S. thunbergii Why is there no quantitative analysis in the manuscript? Please provide the content of active ingredient that can be detected at present (especially the seven active ingredients shown in Figure 1)?

- What are the units of this content? Calculate the absolute quantity

The unit is ug Indole-2-carboxaldehyde/mg extract, as noted in the table footnote. Only the primary active component, Indole-2-carboxaldehyde, was used to validate quantitative data.

  1. The H&E staining in Figure 2 should add a ruler to the figure!

We appreciate your suggestion. We included a ruler in Figure 2.

Reviewer 2 Report

The authors claimed that due to cost, 16S rRNA sequencing was performed on two samples per group by combining the DNA from five out of ten mice in equal proportions in each group. This is not accepatable. I strongly suggest the authors to sequence the gut microbiota individually. Otherwise the authors must further revise the manuscript to specify that this is a limitation of the present study and this could also affect the conclusions of the study.

Author Response

The authors claimed that due to cost, 16S rRNA sequencing was performed on two samples per group by combining the DNA from five out of ten mice in equal proportions in each group. This is not accepatable. I strongly suggest the authors to sequence the gut microbiota individually. Otherwise the authors must further revise the manuscript to specify that this is a limitation of the present study and this could also affect the conclusions of the study.

We've already mentioned it as a limitation of this study, but we've added a few more sentences. In addition, individual DNA from each mouse feces was used for qPCR to augment the data, and the conclusions of the study were made based on the qPCR results rather than the sequencing results (Line 486-490).